# Development and Validation of a Risk Score to Predict Low Birthweight Using Characteristics of the Mother: Analysis from BUNMAP Cohort in Ethiopia

**DOI:** 10.3390/jcm9051587

**Published:** 2020-05-23

**Authors:** Hamid Y. Hassen, Seifu H. Gebreyesus, Bilal S. Endris, Meselech A. Roro, Jean-Pierre Van Geertruyden

**Affiliations:** 1Global Health Institute, Faculty of Medicine and Health Sciences, University of Antwerp, 2160 Antwerp, Belgium; jean-pierre.vangeertruyden@uantwerpen.be; 2Department of Nutrition and Dietetics, School of Public Health, Addis Ababa University, Addis Ababa 1000, Ethiopia; seif_h23@yahoo.com (S.H.G.); bilalshikur10@gmail.com (B.S.E.); 3Department of Reproductive, Family and Population Health, School of Public Health, Addis Ababa University, Addis Ababa 1000, Ethiopia; meselua@yahoo.com

**Keywords:** prediction, model, risk score, low birthweight, pregnant women, decision curve analysis

## Abstract

At least one ultrasound is recommended to predict fetal growth restriction and low birthweight earlier in pregnancy. However, in low-income countries, imaging equipment and trained manpower are scarce. Hence, we developed and validated a model and risk score to predict low birthweight using maternal characteristics during pregnancy, for use in resource limited settings. We developed the model using a prospective cohort of 379 pregnant women in South Ethiopia. A stepwise multivariable analysis was done to develop the prediction model. To improve the clinical utility, we developed a simplified risk score to classify pregnant women at high- or low-risk of low birthweight. The accuracy of the model was evaluated using the area under the receiver operating characteristic curve (AUC) and calibration plot. All accuracy measures were internally validated using the bootstrapping technique. We evaluated the clinical impact of the model using a decision curve analysis across various threshold probabilities. Age at pregnancy, underweight, anemia, height, gravidity, and presence of comorbidity remained in the final multivariable prediction model. The AUC of the model was 0.83 (95% confidence interval: 0.78 to 0.88). The decision curve analysis indicated the model provides a higher net benefit across ranges of threshold probabilities. In general, this study showed the possibility of predicting low birthweight using maternal characteristics during pregnancy. The model could help to identify pregnant women at higher risk of having a low birthweight baby. This feasible prediction model would offer an opportunity to reduce obstetric-related complications, thus improving the overall maternal and child healthcare in low- and middle-income countries.

## 1. Introduction

Low birthweight (LBW), a weight at birth of less than 2500 g (5.5 lb), continues to be a significant public health problem globally. It is estimated that 15% to 20% of all births worldwide are LBW, accounting for more than 20 million in a year [1]. The rate of LBW varies considerably among regions and countries, with higher burden among low- and middle-income countries (LMIC). The prevalence in LMICs (16.5%) is twice higher than in high-income countries (7%) [2]. In Ethiopia, LBW rate ranges from 8% to 54% [3,4,5,6], showing a huge variation across geographical settings and time periods. A recent systematic review showed a pooled estimate of 17.3% in Ethiopia [7], which implies it still remains an important public health problem in the country. 

LBW or being small for gestational age increases infant morbidity and mortality [8,9,10,11,12]. It is related to childhood health outcomes, such as susceptibility to infection, neurological deficits, and lower cognitive skills [13,14,15]. Later in life, it is associated with high blood pressure, diabetes, and coronary heart disease [16,17,18,19,20]. In 2016, the infant mortality rate in Ethiopia was 48 deaths per 1000 live births, of which a significant proportion was attributed to LBW [21].

Demographic factors such as young maternal age, higher birth order, prim-gravida, low educational level, and poor maternal nutritional status before and during pregnancy are well recognized risk factors for LBW [7,22,23,24,25]. Numerous other determinants have also been associated with intrauterine growth retardation, such as rural residence, poor diet, anemia, parity, and presence of chronic illness [25,26,27]. Socioeconomic factors including household income and level of education have also been suggested [26,28]. 

LBW has a remarkable impact on the political, social, economic, and healthcare system in LMICs. Hence, by the end of 2025, the World Health Assembly set a policy target to reduce LBW by 30% [1]. Strategies have been implemented with given emphasis on the packages of care provided at the prenatal, ante-natal, intra-natal, and post-natal period. As a result, the proportion of mothers attending antenatal care (ANC) is improving. As part of the strategy, it is essential to diagnose or predict fetal growth restriction earlier in pregnancy to take appropriate measure for high risk groups. However, in LMICs, imaging equipment and trained manpower are limited. It is assumed that a simple prediction tool could be an alternative in resource-poor settings. However, no significant clinical attempt has been made to predict the probability of LBW. To our knowledge, two studies [29,30] tried to develop a prediction model, although they had less practical implication because the predictors used are not easily obtainable in primary healthcare settings. We developed and validated a model and risk score to predict LBW in primary care settings of LMICs. The risk scores could be used by clinicians and public health professionals working on maternal and child health unit to predict LBW earlier in pregnancy.

## 2. Methods and Materials

### 2.1. Study Setting

The present study used data from the Butajira Nutrition, Mental health and Pregnancy (BUNMAP) project in Ethiopia, a population-based cohort established in 2016. It is a cohort of pregnant women and their offspring living in selected clusters of Butajira Health and Demographic Surveillance Site (HDSS), South Ethiopia. Butajira HDSS is one of the oldest surveillance sites in Africa established in 1986. The livelihood of the residents is based on subsistent farming. Khat (Catha edulis Forsk) and chili peppers are the main cash crops, while maize, banana, and Ensete (Enseteventricosun) are the main staples. The cohort is still ongoing and by the time of this analysis, 881 pregnant women were enrolled and planned to follow the mother-child pair up to the third birthday of the child. Among those enrolled, 388 gave birth, whereas the remaining (493) were in first (245), second (156), or third (92) trimester of pregnancy at the time of this analysis (May 2019). 

### 2.2. Ethical Statement

Ethical clearance was obtained from the Institutional Review Boards (IRB) of Addis Ababa University, College of Health Sciences (code: 099/17/SPH). Written informed consent and parental assent were obtained from study participants after explaining the possible risks, benefits, purposes of the study, issues of confidentiality and voluntarism. The study is in compliance with the principles of the declaration of Helsinki. All data were stored either in password-protected computers or, in the case of paper records, in locked files in the project’s locked office in Addis Ababa, Ethiopia. 

### 2.3. Study Design and Participants

The theoretical design of the present study was; the incidence of low birthweight (at time 1) as a function of multiple predictors during pregnancy (time 0). The source population for the cohort were all 15 to 49 years old women living in Butajira HDSS, who have the capacity to be pregnant. All pregnant women who were enrolled into the cohort and fulfilled the eligibility criteria were included in the analysis. To be included in this study, women must meet all of the following eligibility criteria; (1) should gave birth and (2) birthweight was taken within 72 h of delivery. Whereas, women with fetuses having congenital malformations during the ultrasonographic evaluation or twins or above pregnancies were excluded. Out of 388 pregnant women who gave birth, for 8 of them birthweight was not taken within 72 h of delivery and 1 delivered for twin babies, making our final sample 379. 

### 2.4. Data Collection

Outcome Assessment. After enrollment, three ultrasound examinations of pregnant women were done, one during each trimester, to estimate gestational age, intrauterine fetal growth, and presence of any congenital anomaly. Birthweight was taken within 72 h of post-delivery using a digital scale. The main outcome, LBW, was defined as a weight of neonate below 2500 g (5.51 pounds).

Predictor assessment. A questionnaire was adapted from the Ethiopian Demographic and Health Survey and other relevant literatures. A range of socio-demographic, obstetric, and clinical characteristics of the women including, morbidity, educational status, marital status, occupation, gravidity, parity, ANC utilization, family planning, and the interval between pregnancies were collected. Nutritional status including height, and weight was taken for all women at baseline and during each trimester of pregnancy. The level of anemia was also assessed by measuring hemoglobin in red blood cells, using a Hemo-Cue (Hb-201) instrument. 

### 2.5. Quality Assurance Mechanisms

Training was given for data collectors and supervisors about the objective of the research, how they will collect the data, keep the collected data, and supervise the data collection process. Afterward, a pilot study was done in order to assure that data collectors and supervisors are competent enough to collect and supervise the data collection process. In the case of paper form, questionnaires were controlled for completeness and logical errors, and where errors were found, the questionnaires were redone. 

### 2.6. Data Processing and Analysis

The data were collected using the Open Data Kit (ODK) platforms and were exported to the R statistical programming language version 3.6.0 for further processing and analysis. There was 8 (2.1%), 7 (1.8%), and 6 (1.6%) missing values for hemoglobin level, weight, and height measurements. Moreover, marital status, alcohol consumption and the presence of chronic morbidity each had 1 (0.3%) missing values (Table 1). We assumed data were missing at random, and we therefore performed a multivariate imputation by chained equations using “mice” package in R [31,32]. Missing results were imputed for all variables evaluated in the prediction model, but not for “low birthweight” as we analyzed only participants for whom birthweight was taken. Sensitivity analysis was performed to assess whether the assumption of missing at random (MAR) is valid, and the results were reasonably comparable (Appendix B). Descriptive statistics including mean, standard deviations (SD), median, inter-quartile range (IQR), percentages, and rates were carried out. Incidence and relative risk for low birthweight were also computed. 

#### 2.6.1. Model Development and Validation

We performed a univariable analysis using logistic regression to obtain insight into the association of each potential determinant with LBW and to select potential predictors for multivariable analysis. We fit all the variables with *p*-value < 0.25 in the univariable analysis to the multivariable model to be more liberal. Afterward, we used a stepwise backward elimination technique with *p*-value < 0.10 for the likelihood ratio test to fit the reduced model. As the pregnant women came from different clusters, individual data were likely to be clustered within the different kebeles, which could affect the association of the predictors with the low birthweight. We accounted for such possible non-random differences within kebeles (clusters) using multilevel logistic regression techniques [33,34]. We used a random effect for the intercept (to adjust for differences in baseline rate of low birthweight per kebele) as well as for each candidate variable (to adjust for differences in the associations between variable and outcome per cluster). However, the multilevel analysis identified nearly the same intercept, coefficient, and confidence intervals as the standard multivariable logistic regression analysis. 

To check for the model accuracy, we computed the area under the ROC curve (discrimination) and calibration plot (calibration) using “classifierplots” and “givitiR” packages of R respectively [35,36]. The AUC value of 0·5 indicates no predictive ability, 0·8 is considered as good, and 1 is perfect. The regression coefficients with their 95% confidence intervals, as well as the AUC, were internally validated using the bootstrapping technique [37]. To this end, 2000 random bootstrap samples with replacement were drawn from the data set with complete data on all predictors. The model’s predictive performance after bootstrapping is considered as the performance that can be expected when the model is applied to future similar populations.

To evaluate the clinical and public health impact of the model, we performed a decision curve analysis (DCA) [38], of standardized net benefit across a range of threshold probabilities (0 to 1). In the DCA, the model was compared against two extreme scenarios; “intervention for all” and “no intervention”. In our case, the intervention considered is referral of high-risk pregnant women to facilities with ultrasound or other imaging services. 

#### 2.6.2. Risk Score Development

To construct an easily applicable low birthweight prediction score, we transformed each coefficient from the model to a rounded number by dividing to the lowest coefficient. The number of points was subsequently rounded to the nearest integer or half-integer. We determined the total score for each individual by assigning the points for each variable present and adding them up. The predicted probability of LBW was presented according to three categories of the risk score for reasons of statistical stability and practical applicability. The categories were arbitrarily chosen with a view to reasonable size of each category as well as public health sensibility. Later, the score was transformed to a dichotomous “prediction test,” allowing each pregnant woman to be classified as at high or low risk of LBW. We carried out a sensitivity analysis around different cutoff points of 3, 3.5, 4, 4.5, 5, and 5.5. The sensitivity, the specificity, the positive and negative predictive values, and the likelihood ratios of categorized values of the score were calculated (Appendix C).

This study was reported in accordance with the TRIPOD (transparent reporting of a multivariable prediction model for individual prognosis or diagnosis) statement [39], which included a 22-item checklist to give guidance for reporting the development and validation of a prediction model (Appendix A).

## 3. Results

### 3.1. Baseline Demographic, Obstetric, and Clinical Characteristics of Pregnant Women

We included a total of 379 women who gave birth and birthweight was taken within 72 h of delivery. Table 1 shows the demographic, obstetric, and clinical characteristics of pregnant women included in the analysis. The median age of the mothers was 28 years (IQR: 22–35; and 51 (13.5%) were less than 20 years old). Most (92.6%) of them were married, and 221 (58.3%) never attended any formal education. Above one-third (37.2%) were primigravid, of which above two-thirds (71.0%) have attended at least one ANC visit in their previous pregnancy. A quarter (25.3%) of pregnancies were unplanned, and 127 (33.5%) used family planning before current pregnancy. One hundred four (28.0%) had body mass index (BMI) < 18.5, and 156 (41.8%) were shorter than 155 cm height. Sixteen (4.2%) have history of chronic co-morbidity either cardiovascular, pulmonary, diabetes, or chronic kidney diseases. The hemoglobin test result indicated, 132 (35.6%) had hemoglobin level less than 11gm/dL. Fifty-three (14.0%) of them reported they took alcohol at least once a week.

### 3.2. A Prediction Model for Low Birthweight

Out of 379 women who gave birth, 83 (21.9%) were low birthweight infants. The mean birthweight was 2788.4 g (SD: 611.4). After review of literature, 13 demographic, obstetric, and clinical characteristics of the mother were collected at baseline and considered to predict low birthweight at term. The univariable analysis found several factors were eligible to be included in the prediction model. Variables with *P* < 0.25 in the univariable analysis were; age at current pregnancy, BMI, height, educational status, hemoglobin level, attending previous ANC, gravidity, and presence of comorbidity. Then, six predictors remained in the reduced multivariable regression analysis; younger age (<20 years), underweight (BMI < 18.5), short stature (height < 155cm), anemia (hemoglobin < 11mg/dl), primi-gravida, and presence of comorbidity. Using the results, a prediction model was developed and equation for the prediction model was obtained. (Table 2)

The AUC of the final reduced model was 0.83 (95% confidence interval: 0.78–0.88) (Figure 1a). The calibration test had a *p*-value of 0.89, indicating that the model does not misrepresent the data (Figure 1b). Validation of the model with the bootstrap technique showed hardly any indication of undue influence by particular observations, with optimism coefficient of 0.0092, resulting AUC of 0.82 (corrected 95% CI: 0.76–0.89). 

Using the coefficients (β) the predicted risk cutoff point was a probability of > 0.2631, with sensitivity of 71% (95%CI: 60–81), specificity 82% (95%CI: 77–86), positive predictive value 52% (95%CI: 43–62), and negative predictive value of 91% (95%CI: 87–94). The positive and negative likelihood ratios were 3.9 (95%CI: 2.95–5.14) and 0.35 (95%CI: 0.25–0.50), respectively. 

As shown in Figure 2, the model has the highest net benefit across the entire range of threshold probabilities, which clearly indicates that the model has the highest clinical and public health value. Hence, referral decision made using the model has a higher net benefit than not referring at all or referring all regardless of their risk threshold.

### 3.3. Risk Classification Using a Simplified Risk Score

For practical utility, we developed a simplified risk score from the model. Rounding of all regression coefficients in the reduced model resulted in a simplified prediction score presented in Table 2. The simplified score had a considerably comparable prediction accuracy with the original β coefficients, with an AUC of 0.82 (95%CI: 0.76–0.89). The possible minimum and maximum scores a woman can have are 0 and 12.5, respectively. The proportion of LBW were 7.7%, 36.3%, and 73.8%, respectively, in low (score < 4), intermediate (4 to 6), and high-risk group (≥6). (Table 3)

When dichotomized to high risk (>4) and low risk (≥4) based on the risk score, 114 (30.1%) were categorized as high risk and 265 (69.9%) as low risk for LBW. Using “Youden index”, the suggested cutoff to predict LBW using risk scores is > 4 with a sensitivity of 72.3% (95%CI: 61–82), specificity of 81.8% (95%CI: 77–86), positive predictive value of 52.6% (95%CI: 43–62), negative predictive value of 91.3% (95%CI: 87–94), positive likelihood ratio of 3.96 (95%CI: 3.01–5.22), and negative likelihood ratio of 0.34 (95%CI: 0.24–0.48). Detailed information on the risk score performance at different possible cutoff points is available in Annex 2.

## 4. Discussion

The present study shows the incidence of low birthweight was 21.9%. The optimal combination of maternal characteristics to predict LBW are age < 20, BMI < 18.5, hemoglobin < 11 mg/dl, height < 155cm, prim-gravida, and presence of comorbidity. This study quantified the predictive performance of a model using maternal characteristics during pregnancy without any advanced laboratory or imaging tests. 

Predicting the probability of LBW in pregnant women is essential to take appropriate measures accordingly. The WHO recommends one ultrasound for every pregnant women before 24 weeks of gestation to estimate gestational age, fetal weight, and any fetal anomalies [40]. Nevertheless, in LMICs, imaging equipment and trained professionals are merely available in low level healthcare system. Previously, the focus of research was to explain the maternal and fetal determinants of LBW. In recent years, the focus shifted to predicting low birthweight optimally using a combined set of characteristics. In our study, a combination of 6 maternal characteristics results in AUC of 0.83, which is good accuracy according to diagnostic accuracy classification [41]. A study by Singh and his colleagues developed a model using inadequate weight gain by the mother during pregnancy (<8.9 kg), inadequate proteins in diet (<47 g/d), previous preterm baby, previous LBW baby, anemic mother, and passive smoking with a AUC of 0.79 [29]. However, some of the predictors they used such as inadequate weight gain during pregnancy and inadequate proteins in diet are not easily obtainable information in routine clinical and public health practice, which makes the model less practical. On the other hand, Rejali and his associates performed a decision curve analysis involving 15 predictor variables and found a net benefit (NB) of 0.311 [30]. Nevertheless, 4 of the variables included in the prediction model were obtained from factor analysis, reduced from other several variables. Despite its good accuracy, since it demands advanced statistical skill by end users, it is unlikely to be used by health care professionals in routine clinical practice. Our prediction model constitutes variables that are easily obtainable and have reasonable accuracy to be used by both mid- and lower-level health professionals in the primary care settings. Among the maternal characteristics included in our model, 3 can be easily found from history taking, 2 by physical measurements, and 1 by test for hemoglobin using field Hemo-Cue instrument. 

In our prediction score, using 4 as cutoff point has an acceptable level of specificity, sensitivity, PPV, and NPV to predict LBW. It is also possible to shift the cutoff point to increase either of the accuracy measures depending on the program aim and availability of resources. Although the ultrasonographic evaluation of pregnant women gives a better indicator of fetal growth and prediction of birthweight, maternal characteristics during pregnancy alone enabled to predict the risk of low birthweight in advance. Our prediction model is not a replacement for the ultrasonographic assessment of pregnant women; however, it will be a screening tool in resource-poor settings for further diagnostic workup and management options. The simplified risk score derived from the regression models is easier to use in routine clinical and public health practice than the regression models and has comparable discrimination and calibration.

This study has several strengths. Firstly, we used an adequate number of participants with the outcome, i.e., LBW, which helped us to construct the model using a sufficient number of predictor variables. Secondly, we internally validated our model using bootstrapping technique and resulted small optimism coefficient, indicating our model is less sample dependent. Thirdly, our prediction model is constructed from easily obtainable maternal characteristics that make it applicable in primary care settings. However, the findings from this study should be interpreted with the perspective of the following limitations. As a single site study, it is confined to a single area, which needs external validation before using it in another context. Due to small sample size, we did not validate the model in separate datasets. However, the bootstrapping showed minimal optimism, indicating a stable predictive capability of the model. Lastly, since the data were from research setting, where training was given to data collectors, some deviation in data quality was expected in real-world practice. Nevertheless, the predictors included in the model are easy to measure, which indicates the impact on the accuracy of the model in clinical or public health practice is minimal. The model will provide its maximum benefit provided that all the required predictor information is collected. 

## 5. Implications for Practice and Conclusions

This study shows the possibility of predicting LBW using a simple prediction model constructed from maternal characteristics. The prediction score will help to do a risk stratification of pregnant women and to identify those at higher risk of having an LBW baby. Subsequently, high-risk groups can be linked to a center, which is equipped with ultrasound facilities for further assessment and better management during pregnancy, delivery, and post-natal period. Hence, this feasible prediction score would offer an opportunity to reduce neonatal complications related with low birthweight and thus improving the overall maternal and child healthcare. We strongly recommend validating the prediction tool in another context before introducing it to the clinical and public health practices, preferably using real-world data.

## Figures and Tables

**Figure 1 jcm-09-01587-f001:**
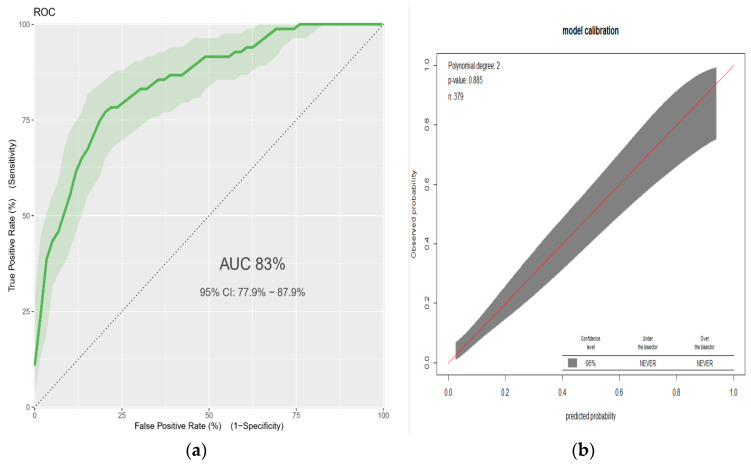
(**a**) Area under the ROC curve for the prediction model, and (**b**) predicted versus observed low birthweight probability in the sample. This analysis includes neonates born at term (*n* = 379). The calibration plot created using “*givitiCalibrationBelt*” in R programming. Linear predictors for estimated risk of low birthweight = 1/(1 + exp − (−2.54 + 1.593 × age(<20) + 1.516 × BMI (<18.5) + 1.213 × hemoglobin (<11) + 1.225 * height (<155) + 0.606 × prim-gravid + 1.475 × comorbidity. ROC = receiver operating characteristic.

**Figure 2 jcm-09-01587-f002:**
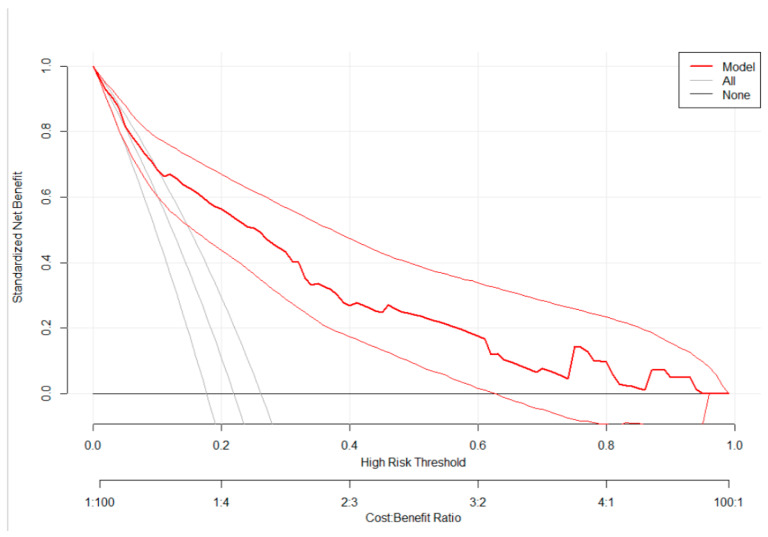
A decision curve plotting net benefit of the model against threshold probability and corresponding cost-benefit ratio.

**Table 1 jcm-09-01587-t001:** Baseline demographic, obstetric, and clinical characteristics of pregnant women who were enrolled in the Butajira Nutrition, Mental health and Pregnancy (BUNMAP) project, south Ethiopia, 2016–2019 (*n* = 379).

Characteristics	Missing	Frequency	Percent
Age (<20 years)	0 (0.0)	51	(13.5)
Marital status (without partner)	1 (0.1)	28	7.4
Formal education (no)	0 (0.0)	221	58.3
Gravidity (prim-gravida)	0 (0.0)	141	37.2
Previous ANC (No) (*n* = 238)	0 (0.0)	69	29.0
Intention to pregnancy (un-planned)	0 (0.0)	96	25.3
Previous Family planning use (yes)	0 (0.0)	127	33.5
Birth interval (<24 months) (*n* = 238)	0 (0.0)	98	41.2
BMI (<18.5)	7 (1.8)	104	28.0
Height (in cm) (<155)	6 (1.6)	156	41.8
Hemoglobin (mg/dl) (<11)	8 (2.1)	132	35.6
Chronic morbidity (yes)	1 (0.1)	16	4.2
Alcohol (at least once/week) (yes)	1 (0.1)	53	14.0
Total		379	100

ANC: Antenatal care; BMI: Body Mass Index.

**Table 2 jcm-09-01587-t002:** Coefficients and risk-scores of each predictor included in the model to predict low birthweight (*n* = 379). Figures are numbers (percentages) unless mentioned otherwise.

Predictor Variable	Univariable Analysis		Multivariable Analysis		Simplified Risk Score
	β (95 % CI)	*P*-Value	β (95 % CI)	*P*-Value	
Age of the mother (<20)	1.596 (0.980, 2.222)	<0.01 ^¥^	1.593 (0.856, 2.344)	<0.01 *	2.5
Marital status (single)	0.184 (−0.779, 1.033)	0.69	NA	-	-
Formal education (no)	0.431 (−0.072, 0.951)	0.098 ^¥^	0.479 (−0.382, 1.384)	0.284	
BMI (<18.5)	1.530 (1.015, 2.053)	<0.01 ^¥^	1.516 (0.915, 2.133)	<0.01 *	2.5
Height (<155cm)	1.032 (0.535, 1.543)	<0.01 ^¥^	1.225 (0.637, 1.838)	<0.01 *	2
Hemoglobin (<11.0 mg/dl)	1.270 (0.768, 1.783)	<0.01 ^¥^	1.213 (0.626, 1.815)	<0.01 *	2
Gravidity (prim-gravida)	0.586 (0.091, 1.080)	0.02 ^¥^	0.606 (0.001, 1.215)	0.049 *	1
Previous ANC (no)	−0.419 (−1.101, 0.288)	0.235 ^¥^	0.011 (−0.885, 0.949)	0.98	-
Birth interval (<24month)	−0.014 (−0.691, 0.647)	0.97	NA		-
Pregnancy (Unplanned)	0.079 (−0.491, 0.622)	0.78	NA		-
Family planning use (yes)	−0.127 (−0.661, 0.387)	0.63	NA		-
Comorbidity ^§^ (yes)	1.627 (0.608, 2.686)	<0.01 ^¥^	1.475 (0.260, 2.744)	0.02*	2.5
Alcohol consumption	0.169 (−0.543, 0.826)	0.63	NA		-

***** Variables retained in the reduced model using likelihood ratio test are; age, BMI, hemoglobin, height, gravidity, and comorbidity. Both backward and forward selection showed same results. β after internal validation with bootstrapping are shown. ^¥^ Variables included in the multivariable analysis (*P* < 0.25 in univariable analysis). § Comorbidity include pulmonary: history of asthma or COPD (chronic obstructive pulmonary disease); cardiac: history of heart failure or ischemic heart disease; renal diseases. NA—not included in the multivariable analysis (*P* > 0.25 in the univariate analysis). Simplified risk score: we divided the coefficient of predictors included in the reduced model by the smallest (0.606). BMI: Body Mass Index; ANC: Antenatal care.

**Table 3 jcm-09-01587-t003:** Risk classification of low birthweight using simplified prediction score (*n* = 379).

Score * (Risk Category)	Prediction Model Based on Maternal Characteristics	
	Number of Women	Incidence of LBW
Low (<4)	246 (64.9%)	19 (7.7%)
Intermediate (4 to 6)	91 (24.0%)	33 (36.3%)
High (≥6)	42 (11.1%)	31 (73.8%)
Total	379 (100%)	83 (21.9%)

* Score = (age < 20 *2.5) + (BMI < 18.5*2.5) + (hemoglobin < 11 mg/dl*2) + (height < 155 cm*2) + (prim-gravid*1) + (presence of chronic morbidity*2.5).

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
