# Peer review of "Development and Validation of a Risk Score to Predict Low Birthweight Using Characteristics of the Mother: Analysis from BUNMAP Cohort in Ethiopia"

_jcm, 2020, doi:10.3390/jcm9051587_

Round 1
Reviewer 1 Report
This article describes creation and validation of a model predicting risk for low birth weight infants using data available in most clinical settings, in a sample of mothers from Uganda. This is an interesting and clearly explained article that provides a practical solution to a maternal-child health challenge in LMICs. Specific comments are below.
First, the authors indicate that the full cohort consisted of 881 women at time of writing, but only 379 women had given birth and were included in analyses. Given that an acknowledged limitation of the analyses was that there was no validation sample for the model, could the authors please justify their decision to create a model using less than half of the whole cohort? Especially given that use of the full cohort could provide a second, validation sample to test model performance, and this would help establish the utility of their model.
Second, regardless, the authors need to provide information on how the partial sample of 379 women compares to the full sample. It is valuable to know how the two pools of participants compare. Also, how many women were excluded because infant weight was not obtained within 72 hours of delivery (or how common is this issue)? Did these women differ in any meaningful way from the women included in the full sample? What are the implications of any differences?
Third, the authors note that the model they test and create is useful because it can be applied in a “real-world” setting. It is possible, however, that some of the protocols and procedures could have reduced the comparability between data collected and data normally available in the “real world.” For example, the researchers ensured that clinicians were well trained in data collection. Do the authors have a sense for how valid or reliable typical clinical data collection is, or at least compared to research data collection? If there are concerns about the accuracy of data collected as part of normal clinical practice, how would that impact their model’s predictive accuracy in a real-world setting?
Fourth, please check the values presented in Table 3. It looks like the total number of women in a risk group (Column 2) is lower than the total number of women with LBW in that category (Column 3).
Minor comment, but there are small grammatical errors throughout the manuscript. For example, on p. 3 a sentence reads “included to the analyses,” which should read “included in the analyses.” Please review for typos and errors.
The authors also use the acronym ANC without, it appears, defining it first. Please define.
Author Response
Response to Reviewers
JCM-777974
Development and validation of a risk score to predict low birthweight using characteristics of the mother: Analysis from BUNMAP cohort in Ethiopia
Journal of Clinical Medicine
I, on behalf of all authors, thank you for your valuable and constructive comments to improve the excellence of this paper. I revised the manuscript based on the comments from the editor and reviewers. Here is a point by point response to the comments and questions.
Reviewer 1
This article describes creation and validation of a model predicting risk for low birth weight infants using data available in most clinical settings, in a sample of mothers from Ethiopia. This is an interesting and clearly explained article that provides a practical solution to a maternal-child health challenge in LMICs. Specific comments are below.
Response: Thank you for your feedback and comments to improve the manuscript. We hope that, we addressed your comments in the revised manuscript. We summarized the point by point response below.
Point 1: First, the authors indicate that the full cohort consisted of 881 women at time of writing, but only 379 women had given birth and were included in analyses. Given that an acknowledged limitation of the analyses was that there was no validation sample for the model, could the authors please justify their decision to create a model using less than half of the whole cohort? Especially given that use of the full cohort could provide a second, validation sample to test model performance, and this would help establish the utility of their model.
Response 1: We thank you for your concern. The bigger cohort has several objectives and aimed to follow pregnant women and their offspring until the third birthday of the child. This specific study is a nested from the big cohort, though the scientific time is prospective. The main reason is an administrative issue. We couldn’t wait until all the mothers gave birth, as it might take months. The big cohort was ongoing and enrolling pregnant women, even after this analysis has been performed. By the time we did the present analysis, 881 were enrolled, of which 388 mothers gave birth, and 245, 156, and 92 respectively were in 1st, 2nd, and 3rd trimester of pregnancy. Our main study populations for current analysis were those who gave birth. Of those who gave birth, 8 of them did not have birthweight measurement within 72 hrs and 1 was twin. The eligible sample (379) resulted 83 LBW, which was sufficient to develop and validate the model, based on 1 to 10 principle as suggested by Grobbee DE. Our sample could accommodate at least 8 predictors, which was sufficient to develop the model, which has 6 predictor variables.
We agree that the larger sample provide more stable prediction model and validating using separate dataset could be useful. However, considering the feasibility and administrative issues, this study also provided unbiased estimate. Since we performed an internal validation using bootstrapping (2000 bootstrap samples), we still believe the model does not miss represent the whole pregnant women in the bigger cohort. Bootstrapping is considered as the most efficient method of internal validation. As per your suggestion, we added more description about the big cohort and the subset cohort in page 2 line 76-79 and page 3 line 95-97 of the revised version. We also provided comments about bootstrapping result in line 290-291 of the revised manuscript. We will also consider to validate the model further using the remaining samples whenever the cohort is completed.
Point 2: Second, regardless, the authors need to provide information on how the partial sample of 379 women compares to the full sample. It is valuable to know how the two pools of participants compare. Also, how many women were excluded because infant weight was not obtained within 72 hours of delivery (or how common is this issue)? Did these women differ in any meaningful way from the women included in the full sample? What are the implications of any differences?
Response 2: Thank you for your feedback. We agree that providing more information on the big cohort and the sample used in this analysis. As we mentioned in point 1, out of 388 who were enrolled, 8 of them gave birth but birthweight was not taken and 1 was twin. The rate of being no birthweight obtained is only 2%, which is very low. For more clarification, we added this description in page 2 line 77-78 of the revised manuscript. As the 388 were a random samples of the cohort (natural random), we believe that there is less likelihood of systematic difference between those who gave birth and those who did not. In the preliminary analysis, we checked for the baseline sociodemographic characteristics, no significant variation was observed. However, we could not ascertain any deviation in the outcome, as it is not measured yet. Nevertheless, we do not expect much deviation as there is no systematic selection of the sub-set cohort, we considered all who gave birth.
Point 3: Third, the authors note that the model they test and create is useful because it can be applied in a “real-world” setting. It is possible, however, that some of the protocols and procedures could have reduced the comparability between data collected and data normally available in the “real world.” For example, the researchers ensured that clinicians were well trained in data collection. Do the authors have a sense for how valid or reliable typical clinical data collection is, or at least compared to research data collection? If there are concerns about the accuracy of data collected as part of normal clinical practice, how would that impact their model’s predictive accuracy in a real-world setting?
Response 3: Thank you for your question. We agree that there could be some deviations in the quality of data collection in research and real-life setting. We added a comment on it in limitation section, line 294-297 of the revised manuscript. In fact, for the outcome data, we need to train them to obtain very accurate data as much as possible. In prediction models, it is recommended to avoid or minimize misclassification of the outcome. For predictors, real-world data is preferable. However, as the variables included in our prediction model are easy to measure, the deviation might be minimal. The model will provide its maximum benefit provided that all the required predictor information are collected. We also recommended validation in real-world data or we might consider it in the follow up project.
Point 4: Fourth, please check the values presented in Table 3. It looks like the total number of women in a risk group (Column 2) is lower than the total number of women with LBW in that category (Column 3).
Response 4: Thank you for your comment. It was editing problem. We revised it accordingly in page 8.
Point 5: Minor comment, but there are small grammatical errors throughout the manuscript. For example, on p. 3 a sentence reads “included to the analyses,” which should read “included in the analyses.” Please review for typos and errors.
Response 5: Thank you for your comment. We revised the whole manuscript for possible grammatical and spelling errors.
Point 6: The authors also use the acronym ANC without, it appears, defining it first. Please define.
Response 6: Thank you for your comment. We write in full form in its first appearance.
Kind regards;
Hamid Y. Hassen (PHD student)
Hamid.Hassen@uantwerpen.be
+32466298748
Reviewer 2 Report
The work is very well designed and as a whole very well structured, clear and concise.
The statistical analysis performed is very well designed and offers very clear results.
I only think that the exposition of these results would be much clearer if a column were added to Table 2 with the percentages of each of the variables with respect to the cases within the total sample of mothers of children with low weight.
Table 1 could also be simplified, since they are variables with two complementary options, it would suffice to place one of the two options, for example, age <20 years 51 (13.5%).
Author Response
Response to Reviewers
JCM-777974
Development and validation of a risk score to predict low birthweight using characteristics of the mother: Analysis from BUNMAP cohort in Ethiopia
Journal of Clinical Medicine
I, on behalf of all authors, thank you for your valuable and constructive comments to improve the excellence of this paper. I revised the manuscript based on the comments from the editor and reviewers. Here is a point by point response to the comments and questions.
Reviewer 2
The work is very well designed and as a whole very well structured, clear and concise.
The statistical analysis performed is very well designed and offers very clear results.
Response: Thank you for reviewing the manuscript and providing comments to improve the manuscript. We hope that we addressed your comments in the revised manuscript. We summarized the point by point response below.
Point 1: I only think that the exposition of these results would be much clearer if a column were added to Table 2 with the percentages of each of the variables with respect to the cases within the total sample of mothers of children with low weight.
Response 1: Thank you for your comment. We agree that adding a column would add more information. We have been discussing about it, some agree to add a column but some do not. Finally, we decided not to add a column and the reason was; providing a percentage of cases for a single category has no more added value unless it compared with the second category. For instance, adding percent of LBW among mothers aged <20 years has no additional information unless compared with those aged >20 years. However, adding the percentage for both categories would complicate the table (more column and rows). As a prediction model, the main interest is the weight contribution of each variable to the risk score, we think that the information given in the table is sufficient not to overload information in the table. If you still think that adding a column is necessary, we will reconsider it again.
Point 2: Table 1 could also be simplified, since they are variables with two complementary options, it would suffice to place one of the two options, for example, age <20 years 51 (13.5%).
Response 2: Thank you for your comment. We revised table 1 in page 5 as suggested. We hope that the table is now simplified without losing any information.
Kind regards;
Hamid Y. Hassen (PHD student)
Hamid.Hassen@uantwerpen.be
+32466298748
Round 2
Reviewer 1 Report
All of my comments have been addressed, except for the one relating to sample size and validation. The authors note that they chose not to re-run models for the larger sample due to "administrative issues" and that "it might take months" for more data to be accessible. This is not a sufficient reason, especially given that the authors have stated that the data will be available shortly.
The authors also mention that this is a sub-study, but do not indicate how their sample differs from the larger sample (e.g. was different data collected, etc.?). If there is no fundamental distinction between their sub-sample and the larger sample, then again there is no justification for running the models only on a smaller sample - especially if more data will be available.
I would strongly advise the authors to re-consider or seriously strengthen their rationale for focusing on only data that is conveniently available from an on-going study. Otherwise, in the interest of quality science and prediction, it would be best for the authors to wait until all data is available before running analyses.
Author Response
Response to Reviewer
JCM-777974
Development and validation of a risk score to predict low birthweight using characteristics of the mother: Analysis from BUNMAP cohort in Ethiopia
Journal of Clinical Medicine
I, on behalf of all authors, thank you for reviewing the manuscript for the second time to improve the manuscript. Here is a point by point response to the comments and questions.
Reviewer 1
All of my comments have been addressed, except for the one relating to sample size and validation. The authors note that they chose not to re-run models for the larger sample due to "administrative issues" and that "it might take months" for more data to be accessible. This is not a sufficient reason, especially given that the authors have stated that the data will be available shortly.
The authors also mention that this is a sub-study, but do not indicate how their sample differs from the larger sample (e.g. was different data collected, etc.?). If there is no fundamental distinction between their sub-sample and the larger sample, then again there is no justification for running the models only on a smaller sample - especially if more data will be available.
I would strongly advise the authors to re-consider or seriously strengthen their rationale for focusing on only data that is conveniently available from an on-going study. Otherwise, in the interest of quality science and prediction, it would be best for the authors to wait until all data is available before running analyses.
Response 1: Thank you for your feedback. First, we apologize for the mis-use of words and phrases in the previous RTR. What we said “administrative issue” is not on cost of scientific validity. It is to explain that the women were not enrolled at the same time and do not have EDD at the same week or month. We agree that larger sample might provide more stable model. However, the sample we used is also sufficient to provide a valid prediction model and the statistical parameters also confirmed that. Let us explain the difference between the big cohort and the sub-cohort. The big cohort was established in 2016 and has several objectives related with maternal nutrition, mental health, and behavioral risks, and maternal and child outcomes. It was ongoing during this analysis and still ongoing. This specific analysis was not part of the aim during conception of the big cohort. HYH and JV discussed the no. of women who could gave birth during 2019 and the statistical power to develop a prediction model. Then, as it had sufficient statistical power, we developed the research question. During the analysis, we had 388 women who gave birth, of which 83 were LBW. Those who gave birth were eligible to be included in the analysis. The sample was sufficient to develop a multivariate prediction model based on the 1 to 10 principle in the lowest category. There is no specific sample size formula for multivariate prediction model. To have stable model, the recommendation is 1 predictor for 10 in the smallest category according to Grobee (DE Grobbee and AW Hoes (2010)). In our case, we have 83 in the smallest category, which can accommodate 8 variables. In our model, we have 6 variables in the reduced model, indicating the sample is more than sufficient.
In general, we understand your concern, however, we believe that our results are still valid for the following reasons. First, the sample size, particularly number with the outcomes, was sufficient to develop the multivariate prediction model as recommended by DE Grobbee (cited above). The calibration plot, along with the test also showed that our model is well calibrated. Second, the sub-cohort are random (natural random) of the big cohort. There is no much difference between the sub-cohort and the big cohort. Therefore, the effect on the weight contribution (risk scores) will be minimal or none at all as the risk scores are rounded integers and half integers. By using more samples, the confidence interval for the coefficient might be narrow but the impact on the value of the coefficient will not be large. It is the coefficient, not the 95%CI, which determine the weight contribution to the risk scores. Third, we performed an internal validation using bootstrapping (2000 bootstrap samples) and the over-optimism coefficient was too small (~0.01), indicating the model is less sample dependent. Even though it is computer-intensive techniques, Bootstrapping is considered as the most efficient method of internal validation (Efron, B., & Tibshirani, R. (1997)). Therefore, we believe that our prediction model is internally valid. As you suggested, in the future, we shall consider validating the model externally using real-life clinical and public health data to enhance practicality.
Sincerely,
Hamid Y. Hassen (PHD student)
Hamid.Hassen@uantwerpen.be
+32466298748